# Sulfonitrocarburizing of High-Speed Steel Cutting Tools: Kinetics and Performances

**DOI:** 10.3390/ma14247779

**Published:** 2021-12-16

**Authors:** Mihai Ovidiu Cojocaru, Mihai Branzei, Sorin Ciuca, Ioana Arina Gherghescu, Mariana Ion, Leontin Nicolae Druga, Cosmin Mihai Cotrut

**Affiliations:** 1Department of Metallic Materials Science, Physical Metallurgy, Faculty of Materials Science and Engineering, University POLITEHNICA of Bucharest, 060042 Bucharest, Romania; mihai.cojocaru@upb.ro (M.O.C.); sorin.ciuca@upb.ro (S.C.); ioana.gherghescu@upb.ro (I.A.G.); mariana.ion@sim.pub.ro (M.I.); cosmin.cotrut@upb.ro (C.M.C.); 2Section IX-Materials Science and Engineering, Technical Sciences Academy of Romania, 030167 Bucharest, Romania; ld@uttis.ro

**Keywords:** sulfonitrocarburizing, high-speed steel (AISI T1, HS18-0-1), kinetics, carbamide (CON_2_H_4_), sulfonitrocarburizing pastes (mixtures), maximum permissible cutting speed, maximum cut length

## Abstract

The scholarly literature records information related to the performance increase of the cutting tools covered by the superficial layers formed “in situ” when applying thermochemical processing. In this context, information is frequently reported on the carbamide role in processes aiming carbon and nitrogen surface saturation. Sulfur, together with these elements adsorbed and diffused in the cutting tools superficial layers, undoubtedly ensures an increase of their operating sustainability. The present paper discusses the process of sulfonitrocarburizing in pulverulent solid media of high-speed tools steel (AISI T1, HS18-0-1) and its consequences. The peculiarity of the considered process is that the source of nitrogen and carbon is mainly carbamide (CON_2_H_4_), which is found in solid powdery mixtures together with components that do not lead to cyan complex formation (non-toxic media), and the sulfur source is native sulfur. The kinetics of the sulfonitrocarburizing process, depending on the carbamide proportion in the powdered solid mixture and the processing temperature, was studied. The consequences of the achieved sulfonitrocarburized layers on the cutting tools’ performance are expressed by the maximum permissible cutting speed and the maximum cut length. An interesting aspect is highlighted, namely the possibility of using chemically active mixtures. Their components, by initiation of the metallothermic reduction reaction, become able to provide both elements of interest and the amount of heat needed for the ultrafast saturation of the targeted metal surfaces.

## 1. Introduction

When operating the cutting tools in an extremely complex state of stress, defined as contact stresses of the order of gigapascals and more, at the cutting edge, the pressure required for the deformation and cutting of the processed material is also at a particularly high level.

Thus, Gheller [1] states that the level of contact stress between the active part of the tool and the workpiece can exceed 4 GPa in the cutting process. Sergheicev and Pecicovski [2], referring to high-speed steels (HSSs), indicate that they have a special behavior during the processing of steels with ultimate strength over 1 GPa, or austenitic steels, which are difficult to process.

Under these conditions, the cutting edge is subjected to multidirectional compression, unevenly distributed, generating tangential stresses, which engender much more plastic-like states for these parts of the tool material when compared to their initial state.

At very high values of these stresses and especially in situations where their presence is accompanied by local heating, the deformation of thin surface areas can be recorded (even for materials with low plasticity, with a martensitic structure and high proportion of carbides, as in the case of HSSs) [1,2,3].

Starting from the premise that between the compression strength and the deformation resistance to small plastic deformations there is a direct dependence relationship, and the compression strength is strictly correlated with the hardness, Dulamita and Gherghescu [3] state that the HSSs tools, designed for high-speed cutting tools, can be used in conditions of maximum safety until the contact temperature rises to 600–650 °C, with the hardness remaining at a value of about 60 HRC. Likewise, Gheller [1], referring to the same aspects, mentions that the hardness of T1 steel at 500 °C is about 60 HRC. Sergheicev and Pecicovski [2] claim that HSSs grades are characterized by high temperature stability and wear resistance up to about 600 °C.

At some distance from the active area of the tools, there are bending stresses and torsional shear stresses.

High-speed steel intended to chip, as a cutting material, is the second most basic material widely used in manufacturing cutting tools after cemented carbide in this market [4,5,6].

Increasing the performance of cutting tools made of these steels can be ensured by saturating their strongly stressed surface layers with elements such as nitrogen, carbon, and sulfur (saturation with single elements or simultaneously with two or even all three elements), possibly by applying thermochemical treatments [7,8,9,10,11,12,13,14,15,16,17,18,19]. Thereby, an increase in the cutting tools’ operating sustainability is augmented up to twice the starting amount. Such processing is especially effective in the case of tool categories that retain the thermochemically induced layer after resharpening. The most common thermochemical treatments operations applicable to cutting tools made of HSSs are low temperature cyanidation, nitriding, nitrocarburizing, sulfurization, sulfocyanidation, and sulfonitrocarburizing.

Roussesau [7] asserts that plasma nitriding at about 480 °C has the effect of forming a diffusion zone without any compound layer or grain boundary precipitation. Stoicanescu et al. [8] also state that low temperature nitrocarburizing is the most efficient way to obtain increased wear resistance. Psyllaki et al. [9] have observed the formation of a wear-resistant compound layer with a thickness of about 10 microns, due to liquid nitrocarburizing after the Tufftriding technique at an operating temperature of 580 °C, which improves the fatigue behavior of the material. To improve cutting tool performance, the deformation capacity of the layer must be modified compared to the substrate. Strahin [10] states in his thesis that ferritic nitrocarburizing provided an increase of 110% in cycles to failure. All these thermochemical treatments are carried out at relatively low temperatures, in the range of tempering temperatures of processed steels, the saturation rates being relatively high. The authors observed an almost linear dependence of the cyanurate layer thickness on the increase in the current intensity at different holding times; the layer thickness practically doubled when the current increased from 1 A to 8 A.

Sulfocyanidations and sulfonitrocarburizing, as the final operations of the thermochemical processing of tools, can be performed in liquid, gaseous, or powdered solid media [1,2,3,20,21,22]. Nitridability, as Toth et al. state in their review [20], depends directly on the alloying elements. However, the nitriding temperature must be lower than the specific tempering temperature of the steel, so that the core is not affected. After nitriding, values of up to 1200 HV can be obtained for the hardness of the layer. Some modern methods, such as electrical stimulation, are applied to speed up the diffusion process, as Todoroc and Giacomelli describe in their paper [21]. Thus, during electrolysis in stationary electric current, the anodic dissolving process takes place. The authors observed an almost linear dependence of the thickness of the cyanate layer on the increase of the current density at different maintenance times; the layer thickness practically doubled when the current increased from 1 A to 8 A.

Mediums used both for ferritic nitrocarburizing (liquid mediums) and for sulfonitrocarburizing (liquid or frequently powdery solid mediums) contain or may generate components with high toxicity (such as cyanides) during heating, because of reactions between components. One variant that could reduce the toxicity of these mediums would be the thermochemical treatment of carbonitration [11,12,13]. In essence, this type of thermochemical processing has many similarities with low temperature cyanizing (ferritic nitrocarburizing), the differences being the replacement, for example, of cyanides with cyanates, which causes a considerable decrease in toxicity. Another way to reduce the level of toxicity is to replace carbonates in the composition of liquid mediums or to substitue powdery solids that contain carbamide and are intended for ferritic nitrocarburizing with chlorides of alkali metals, for example [23]. This creates the possibility of diminution until the formation tendency of toxic components is blocked, without affecting the activity of the mediums.

In the layers obtained by the thermochemical processing of HSS, plasma or ion nitriding, ferritic nitrocarburizing, sulfonitrocarburizing, and complex nitrides of tungsten and chromium are formed, of the type (W,Fe)_2_N and (Cr,Fe)_2_N, as well as carbonitrides of the type Me_23_ (CN)_6_ and Me_3_(CN). Apart from these phases, it is also found that the solid solution α is saturated with nitrogen, raising its hardness. The hardness of HSS, after ferritic nitrocarburizing performed in molten cyanide salt baths (FeCN), rises by 150–300 units on the Vickers scale [2], reaching values of the order of HV1000–1100. If sulfur is present in the thermochemical processing medium, iron sulfides will be present in the areas adjacent to the surface of the layer.

The experimental research carried out by the group of authors aimed at estimating the performances of cutting tools, removable tools made of T1 HSS, hardened in a non-toxic powdery solid medium based on carbamide and alkali metal chlorides, in the range of temperatures related to ferritic nitrocarburizing.

## 2. Materials and Methods

To describe the growth kinetics of the sulfonitrocarburized layer on the high-speed steel T1 (1.3355, HS18-0-1, ISO 4957-02) grade, Φ 15 × 10 mm samples with the chemical composition 0.71% C; 17% W; 1.25% V; 4.08% Cr, 0.31% Mn, 0.51% Si, 1.18% Mo, and 0.055% Co were taken from forged bars and annealed, and pure technical iron foil (Fe-ARMCO) 0.1 mm thick were used.

The components, their origin, and the composition of the powdery solid medium used for sulfonitrocarburizing, are as follows:Carbamide (produced by AZOMURES, Targu Mures, Romania, with particle sizes of 2 ÷ 5 mm, with 46% nitrogen content and maximum humidity of 0.3%);Ammonium chloride (analytically pure, purchased through Silver Chemical, Bucharest, Romania);Sulfur powder with particle sizes of 60 ÷ 70 μm and 99.5% purity (Jainson Lab, Mrerut, India);Charcoal powder (SC FANCOM IMPEX SRL, Lupeni, Romania).

The homogenization of the components was performed in a ball mill with a capacity of 2.5 L for 30 min at a mill speed of 102 rpm. The sulfonitrocarburizing process was performed in furnaces with automatic temperature control (UTTIS-SA, Vidra, Romania).

The experimental results were investigated by optical microscopy (OM), using an image analysis system (Buehler Omnimet Enterprise Software, Lake Bluff, IL, USA), by scanning electron microscopy (SEM), using a Phenom ProX SEM (Eindhoven, The Netherlands) equipped with an X-ray spectrometer with EDS (Eindhoven, The Netherlands) energy dispersion and by X-ray diffraction using the Rigaku Smart Lab X-Ray Diffractometer (Wilmington, DC, USA). To identify the phases, the DIFFRAC.EVA v4.2.1 program (Karlsruhe, Germany) with the ICDD 2020 database was used (Newtown Square, NY, USA).

The microhardness of the layers obtained by the sulfonitrocarburizing process under different conditions was estimated by using a CV400 micro-Vickers hardness tester (TECNIMETAL, Madrid, Spain). The hardness values were the average of ten indentations. The indentations were performed at different distances from the surface of the layer.

For metallographic investigations, samples were prepared metallographically and etched with Nital 3% reagent.

To verify the media activity used for sulfonitrocarburizing in a powdered solid medium, Fe-ARMCO foils were thermochemically processed simultaneously with samples of HSS tools. By medium component combustion and analysis of the resulting gases, information was obtained related to their sulfonitrocarburizing capacity. The chosen methods were based on infrared absorption (complying with the requirements of ASTM E 1019-18) and LECO TC-236 apparatus (LECO Corporation, St. Joseph, MO, USA) for determining nitrogen and oxygen contents and LECO-type CS-244 (LECO Corporation, St. Joseph, MO, USA) for measuring carbon and sulfur contents.

The performance of HSS tools in various structural states was agreed to be expressed by means of two indicators:*The maximum permissible cutting speed* (determined by the moment when catastrophic wear occurs);*The maximum cutting length* with the maximum permissible cutting speed (until the tool wears out completely, meaning the maximum cut length between two resharpenings). The test conditions were chosen in accordance with ISO 3685/1993 International Standard—Tool-life testing with single-point turning tools.

To achieve the objectives imposed by the research stage, a removable triangular tool (sides 14 mm long, thickness of 3.5 mm, and a seating angle of 60°), taken from forged bars (kneading 1 ÷ 15, corresponding to a degree of deformation of about 90%) of HSS in which a fine and uniform carbide distribution (3 points according to GOST 19265/T1) was ensured. Forging with a high degree of deformation was performed to ensure fine carbide distribution. This was followed by thermal and thermochemical treatments performed as rigorously as possible. Such tools made of HSS after such a processing flow can successfully replace the sintered hard alloy plates in cutting operations [24].

The use of cutting removable tools is an economical solution in terms of the consumption of HSS, as the tool body does not undergo such complex stress during use compared to its active part. Medvedeva [25] concludes that this part of the tool can be replaced with steel suitable for quenching and subsequent drawing, low or medium alloyed with Cr, Ni, or Mn, which are intended for hot processing. The experimental research was carried out by using a support for the removable tool for external turning, a D clamping type ( model DTGNL2020K16-125 × 25 × 25 mm) (Associated Production Tools Ltd., Glasgow, UK).

The maximum permissible cutting speed was determined by front cutting round bars (180 mm diameter and 600 mm length) made of AISI 5115/16MnCr5 steel grade (EN 1.7131—case hardening steel) in the annealed state (180–200 HB). The maximum cut length was estimated by longitudinally cutting bars at different cutting speeds (less than or equal to the maximum permitted speed estimated in the previous stage).

The determination methodologies used are as follows:

*In the case of front cutting*—for a certain speed (rotation speed of the lathe axis) the cutting starts from the center of the semi-finished product by moving the knife (removable small plate) towards its periphery. This is equivalent to a constant increase in the linear cutting speed, due to the increase in the parameter r in the relationship between the linear cutting speed (V, m/min), rotational speed (n, rpm), and the distance from the center of the blank to the area of interest, r (m). With the help of the microscope mounted on the lathe tool holder, the state of the active part of the knife is constantly monitored, recording the moment and speed at which its catastrophic wear occurs. In this way, it runs at every rotational speed at which it is intended to be operated to estimate the maximum permissible cutting speed.

*In the case of longitudinal cutting*—for a certain cutting speed, the longitudinal semi-finished product is cut, keeping the working parameters constant (cutting speed, feed rate, cutting depth) and consistently following the condition of the blank and the active part of the tool surfaces.

The experiments were conducted as follows:To determine the maximum permissible cutting speed, the feed rate was set at 0.7 mm/rot with a cutting depth of 1.5 mm, without cooling.To determine the maximum chip length, a speed range of 60 ÷ 200 m/min was tested in the conditions of a feed rate of 0.07 mm/rot and a cutting depth of 1.5 mm, without cooling. One may observe that the conditions chosen for testing the tools fall within the value ranges related to surface finishing operations.

The machining experiments were performed on a universal lathe ED 1000 KDIG Holzmann (Holzman-Store, Ilmenau, Germany), equipped with GEMOLITE-type optical microscope (GEM Instruments, Los Angeles, LA, USA), to detect the catastrophic wear moment. The microscope has 10X maximum magnification, with a measurement accuracy of about ± 0.01 mm.

The performance verifications were performed on removable tools whose structural states were determined by the following processing variants:Quenched (austenitizing in neutral melted salt bath BaCl_2_ at 1280 °C for 2 min with oil) and triple-tempered (melted salt bath with NaNO_3_ and NaNO_2_ at 560 °C for 1 h and air cooling for each tempering heat treatment);Quenched, double tempered, and sulfonitrocarburized in powdered solid medium (30% CON_2_H_4_, 10% S, 3.5% NH_4_Cl, and 40% finely divided coal, Al_2_O_3_ balance) under the conditions of 550 °C for 1 h; processing took place in a furnace provided with automatic temperature regulation and control system, produced by UTTIS SA, Vidra, Romania.Quenched, double-tempered, and nitrocarburized in powdered solid medium (30%CON_2_H_4_ and 3.5%NH_4_Cl+40% finely divided coal, Al_2_O_3_ balance under the conditions of 550 °C for 1 h; processing took place in a furnace provided with the automatic regulation system and temperature control, produced by UTTIS SA, Vidra, Romania.Quenched, double-tempered, and nitrided in a gaseous atmosphere (35% NH_3_, dissociation degree), in a furnace with Φ 600 × 900 mm retort dimensions, 35 KW of power, at a temperature of 560 °C for 30 min.Quenched, double-tempered, and ion nitriding at 450 °C for 1.5 h in a gas mixture (NH_3_ and Ar) (p _(NH3+Ar)_ = 2 mbar) in laboratory equipment with 3 KW of power; the batch cooling was performed during the installation in an argon atmosphere of up to 200 °C.

One may observe the following:Hardening and tempering for all the analyzed processing variants were performed in identical conditions from the point of view of thermal and temporal parameters of the chosen media.Thermochemical processing in powdered solid mixtures was performed by placing samples in metallic containers, sealed with metal lids and clay.

## 3. Results and Discussion

### 3.1. Results Regarding the Growth Kinetics of Sulfonitrocarburized Layers on T1 HSS Matrices

The determination of the layer thickness, as well as its final phase composition, is determined in strict correlation with the destination of the tools by the operating stress. Applying thermochemical treatments to the cutting tools obviously leads to an increase in their working performances, as expressed among other parameters by the maximum permissible cutting speed and by the maximum cut length (the cut length between two tool resharpenings).

For example, in the case of gas nitriding applied to ledeburite high-alloy steel, a short-term annealing time is recommended at a temperature of 510 ÷ 520 °C in order to obtain a layer thickness of 10 ÷ 25 μm, as follows: (1) 15 ÷ 20 min for tools with a diameter (or equivalent diameter) below 15 mm; (2) 25 ÷ 35 min for tools with a diameter between 16 mm and 30 mm, respectively; (3) 60 min for massive tools [1].

The current practice in the nitriding field (gaseous or plasma, etc.) aims to choose the thermal, temporal, chemical, electrical, etc., parameters to avoid brittle phase formation on the active parts of the cutting tools. In this context, there is a justified tendency to reduce the temperature and the holding time at the nitriding or nitrocarburizing temperature [26]. For compound layers with thicknesses greater than 2 ÷ 3 μm, the surface fragility of the HSS tools is so great that it causes the active or cutting part of the tool to be damaged [27]. Concerning the tools made of HSS, the thickness of the carbonitride layer must not exceed 1 ÷ 3 μm [27,28].

Previous research has confirmed that carbamide-containing powdered solid media are highly chemically active. To increase the operating performance of ledeburite HSS cutting tools it is necessary, by thermochemical processing in these media, to obtain dimensionally uniform “in situ” layers with a perfectly adherent adequate phase composition within the strictly controllable dimensional limits [23].

In order to achieve rigorous control of the layers’ phase composition, it is necessary to control the phase composition of the chosen media [29]; additionally, in the case of ledeburite high-alloy steels, the processing temperature has to be maintained within the limits imposed by the tempering temperature required by the steel grade, and the annealing time must be within the range of 0.5 ÷ 1.5 h, depending on the thermochemical processing temperature and tool geometry characteristics [26].

Previous research has shown that carbamide proportion increase of over 30% in the media used for the sulfonitrocarburizing of pure technical iron matrices, along with an increase in processing temperature causes a considerable increase in the layer growth kinetics of the three elements. In these areas, nitrogen concentration registers a continuous increase of up to 675 °C in the layer followed by a decrease [23].

In the case of the sulfonitrocarburizing of highly alloyed tool matrices, it has been shown that an increase in the proportion of carbamide may generate an exfoliation tendency in the surface layers.

Consequently, corroborating the information regarding the sulfonitrocarburizing kinetics of pure technical iron in powdered solid mixtures containing carbamide with the requirements imposed on HSS tools, experimental verifications were made on these matrices and in parallel on pure technical iron foils.

The purpose of these verifications was to highlight the effect of the variation of the proportion of carbamide in the powdered solid mixtures CON_2_H_4_-NH_4_Cl-S-C_graphite_-Al_2_O_3_ within the range of 10 ÷ 30% and of the temperature, within the range of 450 ÷ 550 °C, on the size of the sulfonitrocarburized layer, its uniformity, and continuity.

Experimental research has concluded that a variation in the proportion of carbamide within 10 ÷ 30% in powdered solid mixtures containing carbamide along with 10% S, 3.5% NH_4_Cl, 40% C_graphite_, and alumina—balance, are directly reflected in the growth kinetics of the layer and are in strict correlation with the processing temperature (Figure 1).

Thus, at relatively low temperatures, up to 500 °C, the increase in the proportion of carbamide within the range of 10 ÷ 30% does not significantly influence the layer growth kinetics (Figure 2), but at 550 °C the increase in the proportion of carbamide within the range of 10 ÷ 20% leads to an increase of over 50% in the total thickness of the sulfonitrocarburized layer (Figure 3).

One can observe that the process of sulfonitrocarburizing HSS matrices of T1 tools can be carried out efficiently at a temperature equal to its tempering temperature in powdered solid media with lower carbamide contents (20%).

The optical metallography (OM) investigations performed on T1 sulfonitrocarburized steel matrices (Figure 2 and Figure 3) showed the presence of clearly delineated diffusion zones, with dimensions within the range of 10 ÷ 27 μm, as determined by a temperature variation between 450 °C and 550 °C and a proportion of carbamide within the limits of 10 ÷ 30%.

Figure 4 shows some aspects of the Vickers microindentation hardness impression, as well as the values obtained for the three representative areas on the cross-section of the sulfonitrocarburized steel (core area, adjacent to diffusion zone). In this sense, several representative distances were measured, from the surface of the sample, on its section, to the center of the impression (Figure 4b).

Thus, microhardness measurements of the sulfonitrocarburized layer (Figure 4b) showed values up to 1027 HV_0__.__02_ at distances of the order of about 16–30 microns from the surface. These values are in agreement with the data in the literature, regarding the increase in the layer hardness obtained after ferritic nitrocarburizing for this steel’s grade [2]. The microhardness compound layer value was recorded at up to 917 HV_0.02_ at about 3 microns, but not more than 6 microns from the surface; this was a little lower but higher than the microhardness of the areas unaffected by the diffusion of the three elements (about 875 HV_0.02_—core area), which is due to the presence of iron sulfides. The measured values of the microhardness represent the average of 10 measurements. The average measurement made in the core area, at a distance of about 100 μm from the surface, was about 875 HV_0.02_, a value that is very close to that of T1 HSS steel that has been heat-treated and thermochemically unprocessed (hardened and triple tempered).

The presence of sulfur in the thermochemical processing medium in the metal matrix on the surface leads to the formation of a very thin layer of sulfides placed in the area adjacent to the compound layer, followed by an area of carbonitrides. This microstructural morphology has been confirmed by some authors [30,31], who also explains the microhardness variation in the sulfonitrocarburized layer.

Figure 5 presents the X-ray diffraction analysis of the sulfonitrocarburized layers (20% carbamide). The presence of FeC_0.045_ and Fe_7_C_3_ carbides, whose total proportion is about 46%, is highlighted, as it is considerably higher compared to the as-cast or annealed states (about 25%) [1,3], quenched at 1280 °C (about 16%) or quenched and tempered at 560 °C (about 18%), respectively [1].

Nitrides (Fe_4_N, FeN_0.088_), in a total proportion of about 34%, carbonitrides at about 2.6% (Fe_2_C_0.77_N_0.65_), and iron sulfides (FeS, FeS_2_) at about 5.8% were highlighted together with carbides in the sulfonitrocarburized layer.

Figure 6 presents SEM-EDS performed in the layer areas of interest, which revealed increasing variations of carbon concentration from the surface of the sulfonitrocarburized samples towards the layer deeper areas and, respectively, decreasing variations of nitrogen and sulfur. The maximum values of these two elements were registered in the surface proximity and decreased strongly when examined downwards.

The verification of the powdered solid media activity containing various proportions of carbamide intended for sulfonitrocarburizing (Figure 7), was made on 100-micrometer thick Fe-ARMCO foils, substantiating the conclusions regarding the global growth kinetics of the sulfonitrocarburized layer (Figure 1). This discovery was followed by information regarding the change in the medium availability of carbon, nitrogen, and sulfur as determined by the temperature increase within a range of 450 ÷ 550 °C simultaneously with the proportion of carbamide (within a range of 10 ÷ 30%). It was thus found that the greatest change in the sulfonitrocarburizing medium activity is recorded at the concomitant variation of the processing temperature and carbamide proportion, the carbon showing the most spectacular jump, followed by sulfur and finally nitrogen. Thus, as the temperature increased by 100 °C, from 450 to 550 °C, together with an increase in the proportion of carbamide by 20%, from 10% to 30%, the foil average concentration of carbon increased by 400%, sulfur increased by about 156%, and nitrogen increased by 116% (Figure 6). Thus, a significant increase in carbide proportion in the sulfonitrocarburized layer when compared to the one registered in the HSS matrix (doubling their proportion) may be explained.

The goal of applying thermochemical treatments to HSS tools is to ensure their surface hardening without affecting the core characteristics. Achieving this goal becomes possible by enriching the surface layers with elements (C, N) at temperatures not exceeding the tempering temperature, or by correlating the two thermochemical processing parameters (temperature and annealing time), so that no martensite decomposition occurs.

Starting from these premises, superficial zonal hardening was attempted using sulfonitrocarburizing mixtures with specific characteristics of the thermite compositions (Figure 8).

The sulfonitrocarburizing mixtures have components that generate a considerable thermal effect, providing a local, extremely fast metallic surface heating. To initiate these interactions between components, external activation energy is required. Supplying this is possible in many ways, including with the help of a thermal aggregate heated at a minimum of 700 °C, oxygen gas flame concentrated on the mixture surface, laser, etc. The action of the reaction-initiating source should be of short duration to not affect the tool’s deeper zone characteristics that are in the quenched and tempered state. For a mixture composition (14% CON_2_H_4_–41% Fe_3_O_4_–14% TiO_2_–13% Al–4%Mg–14% S), reactions between components are very likely to happen, releasing a sufficiently large amount of energy to ensure carbamide breakdown. Thus, the surface local warming activates the interactions between other medium components and the mass transfer processes from the medium to the metallic matrices surface.

For instance, the equations for the reactions of the following types are:3Fe_3_O_4_ + 9TiO_2_ + 2Al = 9FeTiO_3_ + Al_2_O_3_ΔG_750_
_°C_ = −537 kJ/mol; ΔH_750_
_°C_ = −1937.8 kJ/mol(1)
FeTiO_3_ + 2Al = Fe + Ti + Al_2_O_3_ΔG_750_
_°C_ = −427.6 kJ/mol; ΔH_750_
_°C_ = −364.0 kJ/mol(2)
and generates as a result of their very probable development (ΔG < 0) a thermal effect whose value exceeds 2000 kJ/mol.

X-ray diffraction analysis of T1 sulfonitrocarburized HSS matrices in the mixture (Figure 9) showed that in these special processing conditions, after 1.5 min annealing time at 750 °C, the carbides proportion in the layer had sizes of approximately 2 μm, which is about 26% (close to that of the cast/annealed matrix of T1 steel); that of nitrides was about 4%, and that of carbonitrides was about 2%.

The presence of iron cyanamide was also noted, and the proportion of iron sulfides (FeS + FeS_2_) reached 10%. Because the chemical interaction between components has a short duration, the matrix hardness was not diminished (61 HRC–Figure 8b).

One may conclude that the use of the pastes in order to achieve the sulfonitrocarburization of some areas of interest on the tools made of high-alloy steel tools surface is a method worthy of consideration.

For some paste compositions it becomes possible to achieve surface saturation with the elements of interest even outside the thermal aggregatesrange, simply through the initiation of reactions in short time ranges. It is important that, when choosing these “ingredients” and their proportion, the thermal effect level generated by the reactions between them is taken into account, tending to the highest possible value.

### 3.2. The Performances of Cutting Tools—Removable Plates—Made of T1 Sulfonitrocarburized HSS in Powdered Solid Medium Containing Carbamide, as Compared to the Same Plates Subjected to Other Types of Thermochemical Treatments—Nitrocarburized, Gas Nitrided, Ion Nitrided

Attempts to anticipate the durability of cutting tools exist and are mentioned in the literature [32,33,34]. Thus, both Zajak et al. [32] and Panda et al. [33] determined, through multiple experiments and the statistical processing of experimental data, the mathematical expressions of the correlations between tool durability (T) and cutting speed (Va), using obtained useful information related to the optimal processing speed.

The performances of sulfonitrocarburized tools in a powdery solid medium containing carbamide were assessed with the help of two indicators: the maximum permissible cutting speed and the maximum cut length at a certain cutting speed. Small removable triangular plates were used in order to achieve this goal (Figure 10).

For a better correlation and control of the cutting process, one may consider the relationship between the rotation speed of the lathe mobile shaft (**n**, rot/min), the distance from the cut area to the axis of the semi-finished product subjected to the cutting processing (**r**, m), and the cutting speed (**V**, m/min), as shown in Equation (3).
**V** (m/min) = 2π ∗ **n** ∗ **r**(3)

The graphical expression of this correlation, for speeds within the range of 60 ÷ 400 rpm and distances **r** within the range of 0.05 ÷ 0.085 m (Figure 11), allows the fast estimation of the effects of the change in **r** on the cutting speed **V** when considering a specific rotation speed value, **n**.

When analyzing the graphical expression of the correlation **V** = **f** (**r**, **n**) one may observe that, for a certain value of the rotation of the mobile axis of the lathe, the increase in the distance **r** up to the center of the semi-finished product leads to a change in the cutting speed; the more significant this change, the higher the rotation speed. 

In this way, by front cutting, using removable tools made of T1 steel (Figure 10) in different structural states, the maximum permissible cutting speed (the speed at which the catastrophic wear of the tool occurs) was estimated for each situation (Figure 12 and Figure 13), as follows:

Regime 1—quenched and triple-tempered state;

Regime 2 (SNC)—quenched and double-tempered state at 560 °C for 1 h with sulfonitrocarburized in powdered solid media, composition 20% CON_2_H_4_–3.5% NH_4_Cl–10% S–40%C_graphite_–26.5% Al_2_O_3_; 550 °C for 1 h;

Regime 3 (NC)—quenched and double-tempered state at 560 °C for 1 h with nitrocarburized in powdered solid media, composition 20% CON_2_H_4_–3.5% NH_4_Cl–40%C_graphite_–36.5% Al_2_O_3_; 550 °C for 1 h;

Regime 4 (N gas)—quenched and double-tempered state at 550 °C for 1 h with nitrided in gaseous medium (NH_3_—degree of dissociation 35%) 550 °C for 0.5 h;

Regime 5 (N ionized)—quenched and double-tempered state at 550 °C for 1 h + ion nitrided (450 °C for 1.5 h in a gas mixture (NH_3_ + Ar) (p _(NH3+Ar)_ = 2 mbar).

Figure 13 shows a removable tool treated according to regime 2 (SNC), before use (a) and after catastrophic wear occurs (b). Catastrophic wear occurred in this case, at a cutting speed of about 160 m/min, in a regime without coolant.

From the analysis of the *maximum admissible*, *cutting speed* varies depending on the presence or absence of thermochemical processing subsequent to the standard heat treatment (Figure 12) and their type (if any); one may see that the thermochemical processing applied to the cutting tools when thermal, temporal, and chemical parameters are correctly chosen, ensuring a substantial increase in the maximum permissible cutting speed. The maximum permissible cutting speed increases from 70 m/min to 160 m/min, which signifies a percentage increase of over 140%.

Sulfonitrocarburizing in powdered solid medium, like all processing in powdered solid mediums, does not ensure the possibility of extremely rigorous control of the medium activity or of the processes and phenomena at the medium–surface interface of the metal product, which leads to obtained results that are frequently below the level of those obtained by gas or plasma processing. In fact, the results regarding the maximum admissible cutting speed for the variants with thermochemical processing subsequent to the standard thermal processing are within the variation limits of ± 15%, with a minimum of 143%, which is clearly superior to the non-thermochemically processed ones.

This results in both sulfonitrocarburizing and nitrocarburizing in a pulverulent solid medium containing carbamide represent particularly efficient and economical ways of increasing this indicator of the performance level of the cutting tools.

Regarding the *maximum cut length* at a certain cutting speed (Figure 14), the longitudinal cutting tests with cutting speeds within the range of 70 ÷ 195 m/min was performed. The lower limit corresponds to the HSS standard heat-treated steel tools (hardened and triple tempered), and the superior limit corresponds to the heat-treated and subsequently ion-nitrided tools. The tests highlighted that while for the standard heat-treated tools the maximum cut length was 300 mm at the maximum permissible cutting speed of 70 m/min, the subsequent thermochemical processing of the tools ensured a substantial increase in the maximum cutting length by about 83%, at the maximum permissible cutting speed for thermochemically unprocessed tools.

For tools made of T1 steel sulfonitrocarburized (or nitrocarburized) in powdered solid media containing 20% carbamide, both the maximum permissible cutting speed and the maximum cut length were slightly lower compared to ion-nitrided tools or those in a gaseous environment (the maximum cut length is about 5.5% lower at a cutting speed, which is equal to that of thermochemically unprocessed tools at 70 m/min) but were clearly superior to those provided by thermochemically unprocessed tools.

## 4. Conclusions

Achieving sulfonitrocarburized layers, perfectly adherent and dimensionally uniform and within dimensional limits and phase compositions, as is recommended for T1 HSS grade, using powdered solid medium containing carbamide, is conditioned by limiting the proportion of this organic compound to about 20% in the mixture and the temperature at a maximum of 550 °C.The maximum microhardness recorded in the sulfonitrocarburized layer by T1 steel was 1027 HV_0.02_, at a depth of about 16 ÷ 30 microns; in the surface adjacent area, the presence of iron sulfides leads to microhardness values by about 10.7% lower as compared to those in the deeper zone, where carbonitrides presence was recorded.X-ray diffraction analysis of sulfonitrocarburized layers in a powdered solid medium containing 20% carbamide showed a considerable increase in the carbide proportion when compared to that recorded in T1 steel in as–cast or annealed states (about 25%), quenched (about 16%), or quenched and tempered (about 18%).The activity of the powdered solid medium used for sulfonitrocarburizing changed considerably when the carbamide proportion was increased to 10 ÷ 30% and the temperature was in the range of 450 ÷ 550 °C, the most significant increase being registered for carbon, sulfur, and nitrogen, in descending order.The use of sulfonitrocarburizing mixtures for the controlled zonal modification of the sulfur, carbon, and nitrogen concentrations in the surface layers of HSS represents a special solution from a technical and economic point of view.Sulfonitrocarburizing in a powdery solid medium that contains carbamide of T1 HSS cutting tools ensures a considerable increase in their performance. Thus, there was an increase of over 140% of the maximum permissible cutting speed, while the maximum cut length increased by over 80%, compared to those of thermochemically unprocessed tools working at the maximum permissible cutting speed (about 70 m/min).Sulfonitrocarburizing in a powdered solid medium containing carbamide performs very closely with regards to the cutting tools to thermochemical processing variants such as gas nitriding or plasma nitriding.The proposed process is non-polluting, both reactants and reaction products being environmentally friendly, this being an additional argument, along with the obtained performance, to apply it on an industrial scale.

## Figures and Tables

**Figure 1 materials-14-07779-f001:**
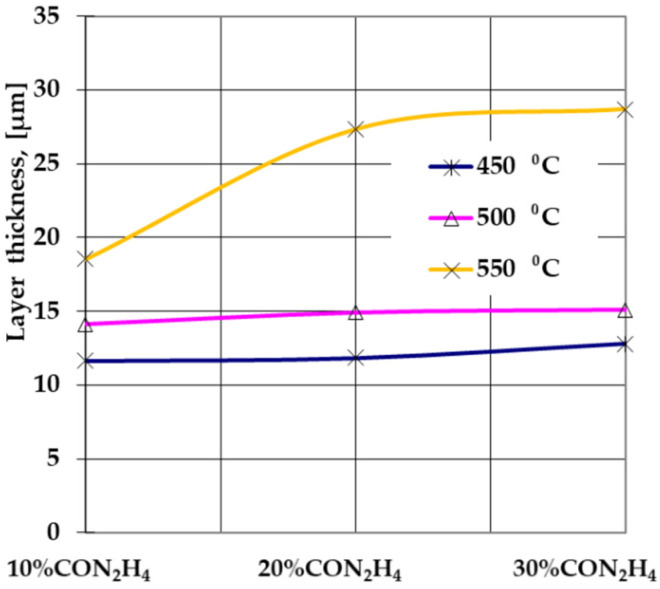
Dependence of the sulfonitrocarburized layer thickness on the proportion of carbamide in the powdered solid mixture: CON_2_H_4_—10% S—3.5% NH_4_Cl—40% C_graphite_-Al_2_O_3_ (balance) and processing temperature; T1 steel; annealing time of 1 h.

**Figure 2 materials-14-07779-f002:**
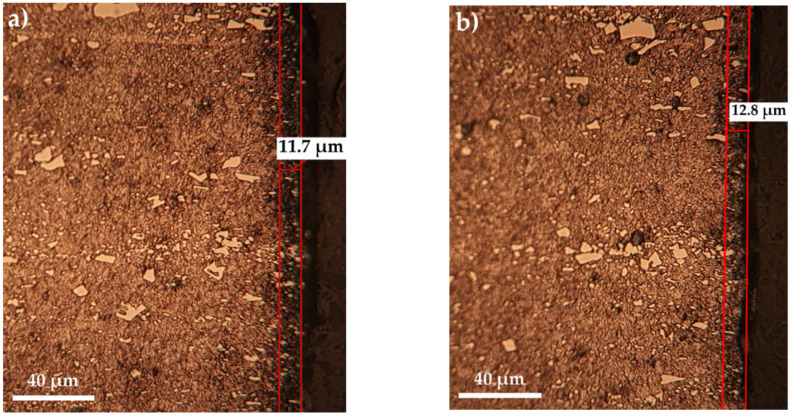
Microstructures of sulfonitrocarburized layers in powdered solid mixtures S-CON_2_H_4_-NH_4_Cl-C_graphite_-Al_2_O_3_ with different proportions of carbamide: (**a**) 10% and (**b**) 30%, respectively, at 450 °C for 1 h, metallic matrix (T1).

**Figure 3 materials-14-07779-f003:**
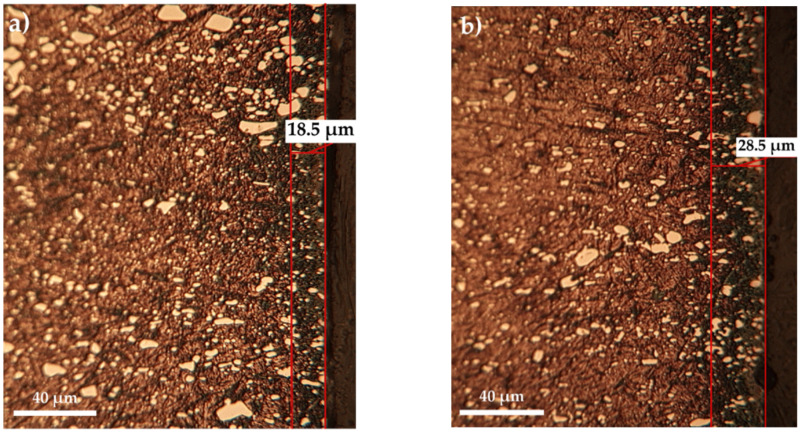
Microstructures of sulfonitrocarburized layers in powdered solid mixtures S-CON_2_H_4_-NH_4_Cl-C_graphite_-Al_2_O_3_ with different proportions of carbamide: (**a**) 10% and (**b**) 20%, respectively, at 550 °C for 1 h, metallic matrix (T1).

**Figure 4 materials-14-07779-f004:**
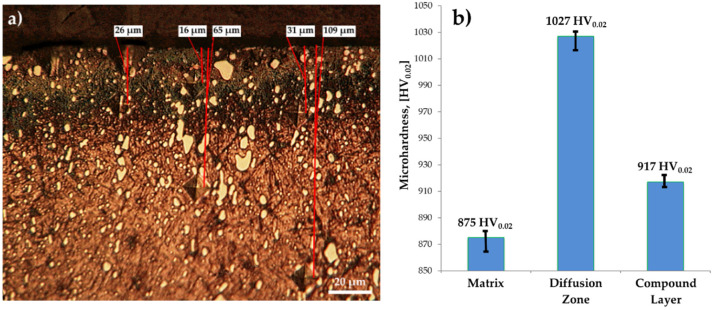
Vickers microindentation hardness impression (**a**) and the average tens measured values of sulfonitrocarburized T1 HSS (**b**).

**Figure 5 materials-14-07779-f005:**
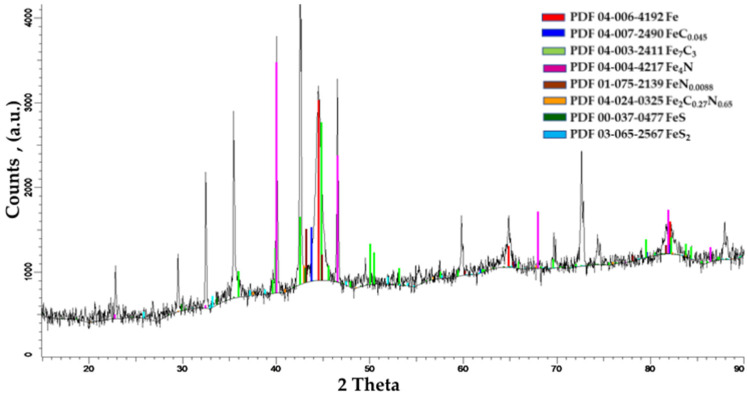
The phase composition of the sulfonitrocarburized layer in the powdered solid medium was determined by X-ray diffraction. Steel T1 grade; medium composition: 20%CON_2_H_4_–NH_4_Cl-S-Graphite-Al2O_3_. Processing conditions: 550 °C for1 h.

**Figure 6 materials-14-07779-f006:**
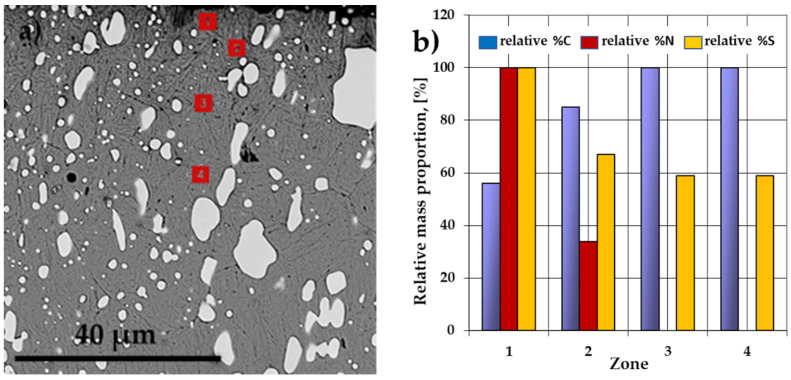
SEM image (**a**) sulfonitrocarburized at 550 °C for 1 h in a powdered mixture of 20% CON_2_H_4_-NH_4_Cl-S-C_graphite_–Al_2_O_3_ with specifications of changes in relative mass proportions (**b**) of the elements of interest in the different investigated zones (1, 2, 3, 4).

**Figure 7 materials-14-07779-f007:**
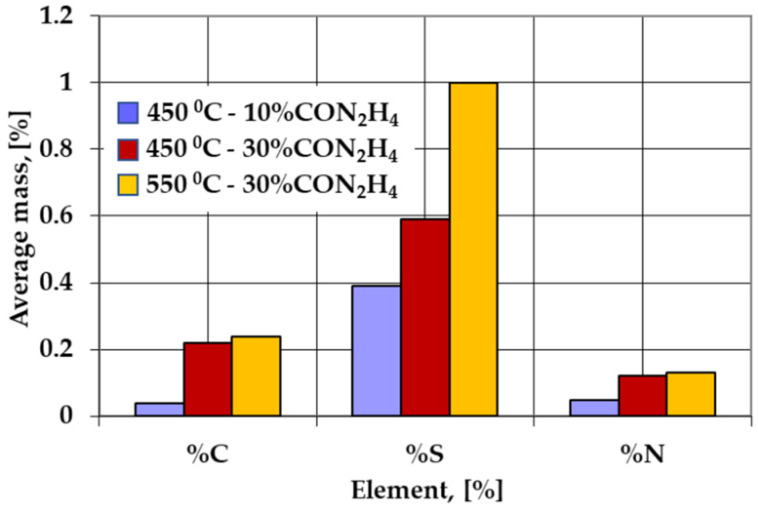
Variations of the average mass proportions of carbon, sulfur, and nitrogen in Fe-ARMCO sulfonitrocarburized foils in the powdered solid medium at various temperatures in the range of 450–550 °C for 1 h, in medium containing various proportions of carbamide (between 10 and 30%). Obs.: Besides carbamide, the medium contains 10%S–3.5%NH_4_Cl—40%Graphite, Al_2_O_3_—balance. The proportions of sulfur, ammonium chloride, and graphite are constant in all recipes, the proportion variation of carbamide within 10–30% being compensated by the proportion variation of alumina.

**Figure 8 materials-14-07779-f008:**
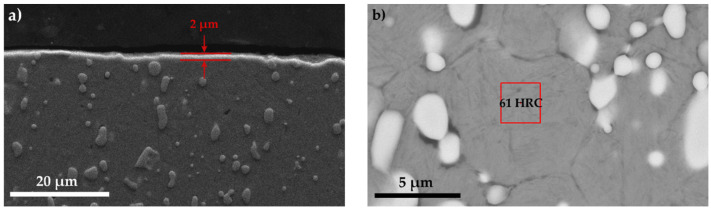
SEM images of the T1 sulfonitrocarburized HSS sample in paste, after 1.5 min in the furnace at a temperature of 750 °C and being cut off. Paste composition: 14%CON_2_H_4_—41% Fe_3_O_4_—14% TiO_2_—13% Al—4% Mg—14% S; polysaccharide resin binder. (**a**) layer area, (**b**) central area of the sample.

**Figure 9 materials-14-07779-f009:**
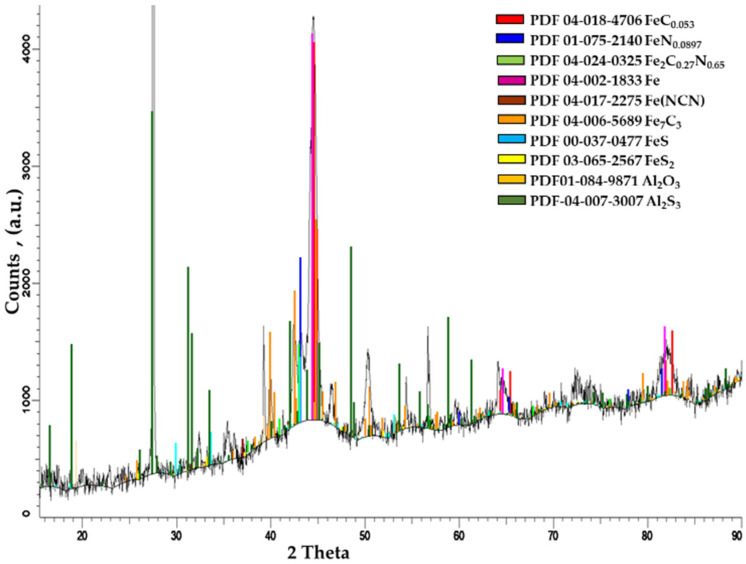
The phase composition of the sulfonitrocarburized layer, determined by X-ray diffraction. T1 steel; Mixture composition: 14% CON_2_H_4_—41% Fe_3_O_4_—4% TiO_2_—13% Al—4%Mg—14% S; polysaccharide resin binder. Processing conditions: furnace temperature at 750 °C for 1.5 min annealing time.

**Figure 10 materials-14-07779-f010:**
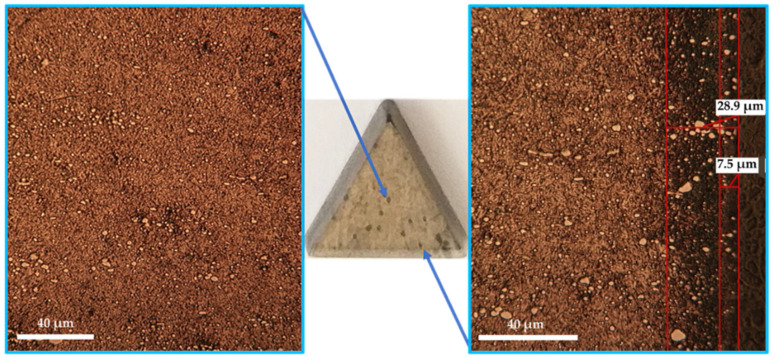
Removable tool of T1 steel and microstructures of characteristic areas: core (quenched and triple tempered) and surface area (thermochemically processed—ion nitrided).

**Figure 11 materials-14-07779-f011:**
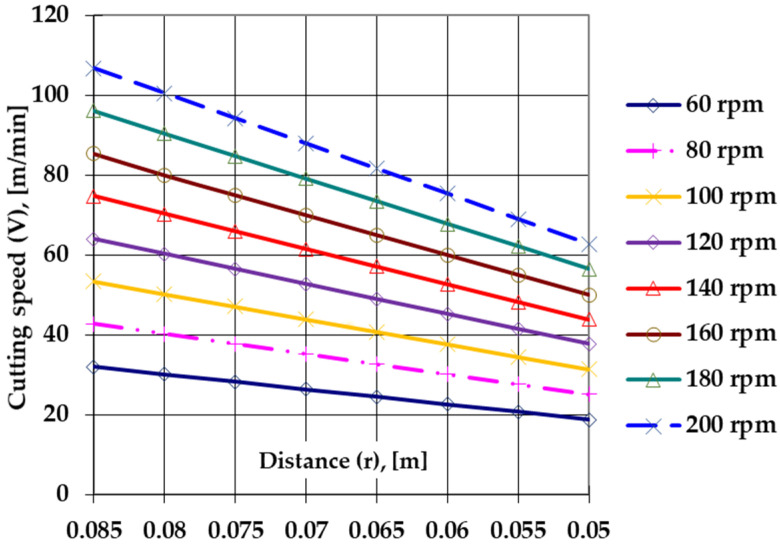
Variation of **V** as a function of **r** for different **n** values.

**Figure 12 materials-14-07779-f012:**
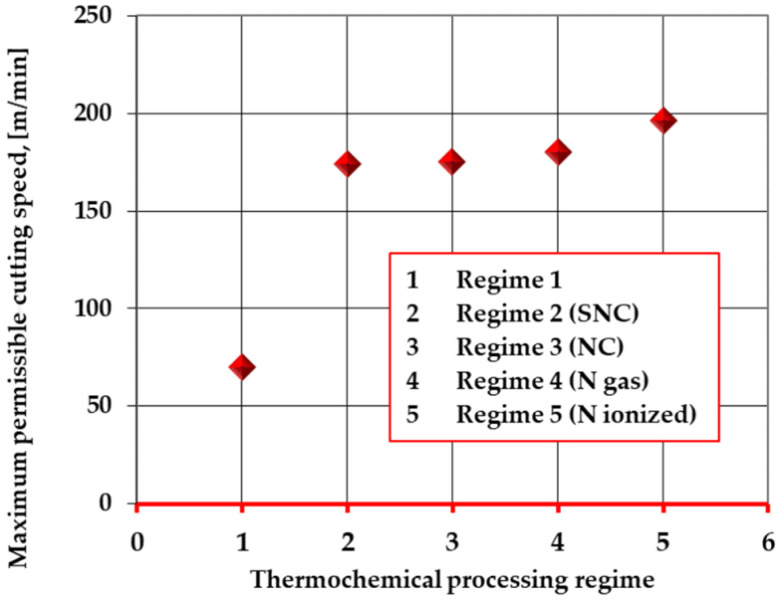
Variation of the maximum permissible cutting speed with the thermochemical processing regime of the steel (T1).

**Figure 13 materials-14-07779-f013:**
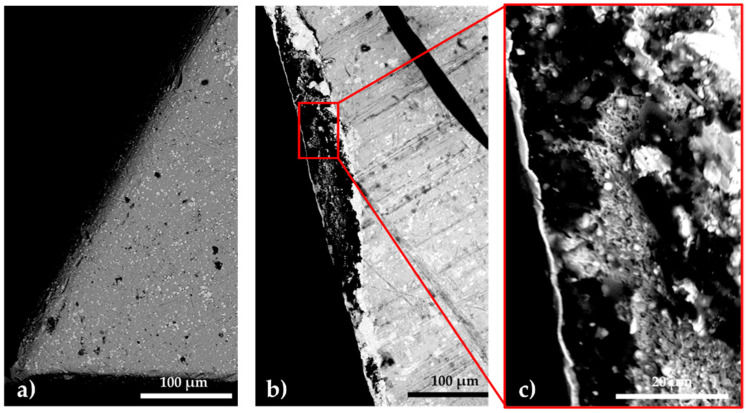
SEM-BSD images of the removable tool active area, before the test (**a**) and after its catastrophic wear occurs (**b**,**c**). The thermochemical processing conditions of the tool are those corresponding to regime 2 (SNC), and catastrophic wear occurred at a cutting speed of about 160 m/min, cutting without coolant.

**Figure 14 materials-14-07779-f014:**
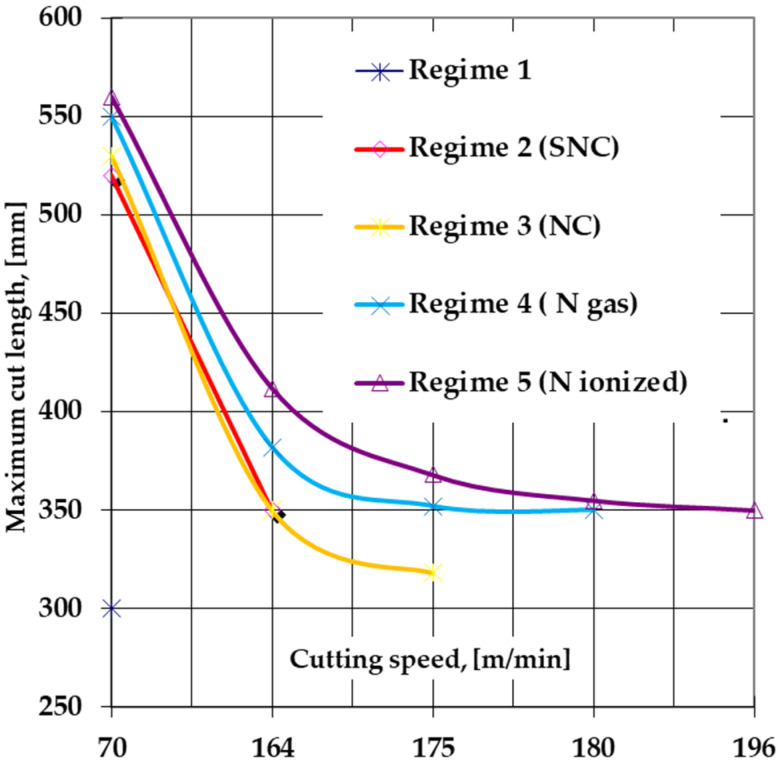
Variation of the maximum cut length before tool resharpening depending on the cutting speed, in strict correlation with the thermochemical processing regime of the tool made of T1 steel. The thermochemical processing regimes are those mentioned in Figure 12.

## Data Availability

The study did not report any data.

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
