# Peer review of "Sulfonitrocarburizing of High-Speed Steel Cutting Tools: Kinetics and Performances"

_materials, 2021, doi:10.3390/ma14247779_

Round 1
Reviewer 1 Report
The article discusses the process of sulfonitrocarburizing in pulverulent solid media of high speed steel tools. The kinetics of the sulfonitrocarburization process, depending on the carbamide proportion in the powdered solid mixture and the processing temperaturę was investigated. The influence of the obtained sulfonitronitroncarbonate layers on the efficiency of cutting tools and their durability was determined.
Sulfonitrocarburizing in carbamide increase performance of cutting tools. There is an increase of over 140% of the maximum permissible cutting speed, while the maximum cut length increases by over 80%, both compared to those of thermochemically unprocessed steel tools working at maximum permissible cutting speed.
The aim of the research specified in the topic was achieved by the authors. The applied research methodology - correct. The article is richly illustrated.
The bibliography concerning the content of the article, in terms of the durability of cutting tools, could be more numerous.
There is little research of this type in world literature, which makes the article unique.
Author Response
Response to Reviewer 1 Comments
Dear Reviewer,
First, I want to thank you for the patience and pertinence with which you reviewed the paper. The observations/suggestions made are to the point and welcome, both for me and for the level of the magazine.
I carefully went through the observations made and the recommended suggestions, trying in the following to answer you in order to them.
Below are presented, in order the changes suggested by you.
General Remarks
- I extended the list of references with 13 references, which are more actual /available/affordable on this topic, as you suggested.
- In order to explain the forms of wear of the cutting tool, I introduced Fig. 13.
- I modified/clarified the problem by measuring the microhardness, introducing a new figure (Fig. 4), with the related explanations.
Finally, I want to thank you once again for the time given and for the equidistance with which you analyzed the paper.
Kind regards,
Mihai Branzei

Reviewer 2 Report
The topic of the article is interesting and also I see the possibility of transferring knowledge directly into industrial practice. The title is adequate to the contribution. The theoretical background is understandable however, in my opinion, list of references should be extended with the latest studies on this topic. A content of the main section is presented clearly. The methodology is sufficient. In the experiment suitable measurement technology was used which contributes to the relevance of the data. The processing of results and conclusions is comprehensible and meets the standards of a scientific article. Supporting figures are appropriate. I consider that the article is suitable for publishing in the MDPI Materials Journal, but some details are missing:
- In Abstract - The abstract should be improved and in the introduction is required to insert some sentences on the state of the problem, in more detail and concrete. What methods are used in the article? What are the quantitative and qualitative results obtained? What is a scientific novelty and practical significance.
- In my opinion Introduction should be improved. I suggest add more information and examples to better describe what other researchers have done in area of the materials and efficiency of cutting tools. Also, chapter "Results and discussion" could be more combined with other published articles and the results achieved by other authors. References should be extended and corrected. The references contain 8 on 20 examples much more than 20 years and only 5 in the last five years.
- Every reference should be analyzed separately and points relevant to the presented paper should be presented. The authors should not use one sentence concerning a few references like in line 64 or 69.
- Fig. 1 - the same values on the horizontal axis.
- In line 137 – feed rate 0.7 mm/rev? - probably should be 0.07 mm/rev.
- Authors should add the description and more details about sample preparation and cutting process condition. There is also no information as to whether the research was conducted in accordance with the adopted research plan. Also, there are some uncertainties regarding cutting tests. What parameters were used to measure tool wear? What were the forms of wear of the cutting tool? How was the wear of the tool during cutting? What was the symbol of the tool holder. What was the chemical composition of the workpiece?
Author Response
Response to Reviewer 2 Comments
Dear Reviewer,
First, I want to thank you for the patience and pertinence with which you reviewed the paper. The observations/suggestions made are to the point and welcome, both for me and for the level of the magazine.
I carefully went through the observations made and the recommended suggestions, trying in the following to answer you in order to them.
Below are presented, in order the changes suggested by you.
General Remarks
- I turned to a Romanian colleague who speaks English, being naturalized in the UK for several years. I modified the unclear and long sentences of the paper, as you suggested.
- I extended the list of references with 13 references, which are more actual /available/affordable on this topic, as you suggested.
- I improved the abstract at your recommendation.
- In the introduction, I inserted some sentences on the state of the problem, in more detail and concrete, as you suggested.
- In Figs. 1 I corrected the values on the abscissa.
- In line 137 I corrected the feed rate 0.7 mm/rev with 0.07 mm/rev.
- In order to explain the forms of wear of the cutting tool, I introduced Fig. 13.
- I modified/clarified the problem by measuring the microhardness, introducing a new figure (Fig. 4), with the related explanations.
- The variation of the chemical composition on the section of the obtained layer is presented in Fig. 6.
References
- I have completed 13 more references, which are more available/affordable.
Finally, I want to thank you once again for the time given and for the equidistance with which you analyzed the paper.
Kind regards,
Mihai Branzei

Reviewer 3 Report
see attached file

Author Response
Response to Reviewer 3 Comments
Dear Reviewer,
First, I want to thank you for the patience and pertinence with which you reviewed the paper. The observations/suggestions made are to the point and welcome, both for me and for the level of the magazine.
Below are presented, in order the changes suggested by you.
General remarks:
- I turned to a Romanian colleague who speaks English, being naturalized in the UK for several years. I modified the unclear and long sentences of the paper, as you indicated on the lines
- I replaced throughout the text the terms used for "high-speed steel material" with the term "high-speed steel (HSS)", as you suggested. I also modified/replaced throughout the text the notation of the steel according to the international standard AISI, as you suggested.
- I have reduced the number of decimals in Fig. 2, 3 9, at a reasonable limit (I left them as they resulted directly from the measurements made with Buehler OmniMet Enterprise software!).
- Regarding the error bars in the mentioned figures, the measurements were performed only once (layer thickness for Fig. 1, X-ray diffraction for Fig. 5, EDS analyzes for Fig. 6, cutting speed for Fig. 11, and maximum permissible cutting speed for Fig. 12), not being possible to repeat the measurements several times, primarily for financial budget reasons. However, these are representative estimated results for the completed grades. Thus, with your permission, please agree that they are representative and comprehensive in the context of the paper.
- I replaced the term "holding time" with "annealing time" throughout the paper, as you suggested.
Introduction
- In your observation, I considered it appropriate to omit the first paragraph.
- I replaced “the cutting part of the tool” with the “cutting edge”, as you suggested.
- I modified the text and introduced the reference mentioned by you (Bobzin, K.).
After the replacement of the micrographs, the paper starts to make sense. This correction was really needed to understand the content well.
Results and Discussion
- I modified the terms in Fig. 1.
- I modified/clarified the problem by measuring the microhardness, introducing a new figure (Fig. 4), with the related explanations.
- I improved the clarity of the images with X-ray diffraction patterns as much as I could (2 Theta is very large) and completed with "a.u." on order.
- I replaced in Eq. 1 and 2 the text “KJ/mol” with “kJ/mol”, as you suggested.
- I also replaced the unit “torr” with “mbar”
References
- I have completed 13 more references, which are more available/affordable.
Finally, I want to thank you once again for the time given and for the equidistance with which you analyzed the paper.
Kind regards,
Mihai Branzei

Round 2
Reviewer 2 Report
Thank you very much. I accept corrections.
Author Response
Dear Reviewer,
First, I want to thank you for the patience and pertinence with which you reviewed the paper. The observations/suggestions made are to the point and welcome, both for me and for the level of the journal.
Finally, I want to thank you once again for the time given and for the equidistance with which you analyzed the paper.
Sincerely,
Mihai Branzei

Reviewer 3 Report
see attached file

Author Response
Dear Reviewer,
First, I want to thank you again for the patience and pertinence with which you reviewed the paper. The observations/suggestions made are to the point and welcome, both for me and for the level of the journal.
Below are presented, in order the changes suggested by you.
General remarks:
- I reformulated, completed, and modified the mentioned sentences and improved the grammar, as you suggested (lines: 44, 56, 79, 85, 221, 227, and 308). I also made corrections to the text.
- I give up the phrase from line 105.
- I replaced the term “divided coal powder” with “charcoal powder”.
Additional remarks:
- I made some additions to the Introduction Chapter.
- I have supplemented and argued in Figure 4 with a comment on the microhardness tests. I would like to inform you that the microhardness tests were also performed to confirm those presented in Figure 6, as they are in accordance with the literature.
Finally, I want to thank you once again for the time given and for the equidistance with which you analyzed the paper.
